# A Narrative Review for a Machine Learning Application in Sports: An Example Based on Injury Forecasting in Soccer

**DOI:** 10.3390/sports10010005

**Published:** 2021-12-24

**Authors:** Alessio Rossi, Luca Pappalardo, Paolo Cintia

**Affiliations:** 1Department of Computer Science, University of Pisa, 56127 Pisa, Italy; paolo.cintia@gmail.com; 2Institute of Information Science and Technologies, National Research Council, 56124 Pisa, Italy; luca.pappalardo@isti.cnr.it

**Keywords:** soccer, artificial intelligence, sport science, training and testing

## Abstract

In the last decade, the number of studies about machine learning algorithms applied to sports, e.g., injury forecasting and athlete performance prediction, have rapidly increased. Due to the number of works and experiments already present in the state-of-the-art regarding machine-learning techniques in sport science, the aim of this narrative review is to provide a guideline describing a correct approach for training, validating, and testing machine learning models to predict events in sports science. The main contribution of this narrative review is to highlight any possible strengths and limitations during all the stages of model development, i.e., training, validation, testing, and interpretation, in order to limit possible errors that could induce misleading results. In particular, this paper shows an example about injury forecaster that provides a description of all the features that could be used to predict injuries, all the possible pre-processing approaches for time series analysis, how to correctly split the dataset to train and test the predictive models, and the importance to explain the decision-making approach of the white and black box models.

## 1. Introduction

With the technological advent of the last few decades, it is possible to record a huge quantity of data from athletes. Wearable devices, video analysis systems, tracking systems, and questionnaires are only a few examples of the devices used currently to record data in sports. These data can be used for scouting, performance analysis, and tactical analysis, but an increased interest is in assessing the risk of injuries. With this huge amount of data, the use of complex models for data analysis is mandatory and, for this reason, machine learning models are increasingly used in sports science. In order to describe the correct methodologies to develop these models, we provide an example focused on injury prediction in soccer.

Injuries have a great impact on the sports industry, affecting both team performance and the club’s economic status. As a matter of fact, injury-related absenteeism from training and matches for top-league players results in a total cost (in terms of player’s recovery, rehabilitation, and players’ salary) of EUR 188 million per season [1]. In addition to soccer, players of other team sports (e.g., rugby, Australian football, and American football) are subjected to a high number of injuries (81 and 6 injuries per 1000 matches and training hours, respectively) [2,3,4]. In contrast, a low number of injuries was detected in kayakers. In fact, only 1.1 injuries/year in men and 1.5 injuries/year in women were detected in this group of athletes [5]. The most common injuries in kayakers are to shoulders, followed by knees, lower back, and the other segments of the upper limbs [5]. Based on these data, it is not surprising that injury forecasting and prevention are becoming prominent topics for researchers, managers, coaches, and athletic trainers.

In the last decade, several papers proposed models to assess athletes’ injury risk. The first injury risk model was developed by Gabbett and colleagues in 2010 [6], who modeled the risk of soft-tissue injury using a monodimensional approach. This risk was estimated through the evaluation of the training load (i.e., the cumulative amount of stress perceived by a player during a single training session) sustained by the athletes during a competitive season. In this study, the authors suggested that soccer players performing a high training workload were 70% more likely to get injured compared to players that were subjected to low training workload. Furthermore, Gabbett, Hulin, and colleagues in the last decade have continued their study by detecting a link between acute training load (i.e., the mean of training load of about one previous week) and injury risk [7,8,9,10]. Essentially, these studies suggested that large increases in acute workload with respect to the chronic workload (i.e., the average training workload of the previous month) are associated with increased injury risk in elite cricket fast bowlers and rugby players. In particular, they showed that players with a high ratio between acute and chronic workload (ACWR 2, i.e., training workload in the last week being 200% higher than the effort that the players used to perform in the last month) are more likely to become injured compared to those with a lower ratio. In particular, they demonstrated that ACWR provides better insights about injury risk than the absolute workload. Additionally, Murray et al. [11] show that aggregating the training load variable (both on the acute and chronic point of view) by using an exponential moving average instead of the rolling mean permits to have a more accurate estimation of the injury risk. However, the use of ACWR model as an injury predictor has been widely criticized in the last years. In particular, Impellizzeri, Kalkhoven, and MacMillan in their last works asserted that no evidence supports the use of ACWR to assess the players’ injury risk [12,13,14]. In accordance with this evidence, Rossi et al. [15] tested the accuracy of ACWR on injury prediction. The results of this study support the criticisms about ACWR on injury prediction showing a low predictive performance of this approach in a real-world scenario.

Since players’ health is affected by several factors linked to the complex human responses to external stimuli, the simplification of the training workload variables into a single feature (e.g., ACWR of one external training workload) does not permit a complete overview of their status. As a matter of fact, this simplification hides the complexity of the training stimuli, not allowing for the detection of complex patterns in training workloads linked to injuries [16,17,18]. For this reason, the literature concerning multidimensional models focused on predicting injury is growing fast [19]. In this review, the authors state that the performance of prediction models was still poor due to the difficulty to solve this very complex modeling problem. One of the main issues is the data sample distribution. Essentially, the injury-related datasets are not balanced among injury and non-injury observations. In essence, it was found that in only 2% of the cases the players incurred in an injury, while the remaining 98% are no-injury examples. This highly unbalanced dataset makes the training of machine learning models difficult in being able to clearly detect workload patterns able to discriminate between the injury and no-injury examples. Oversampling and undersampling approaches could solve this problem, but they were used by only a few papers [15,20,21,22,23]. The aim of these sampling strategies is to balance the dataset allowing highlighting patterns in the training set and consequently enabling the machine learning model to achieve better predictive results.

Testing the predictive performance of the injury prediction models can hide pitfalls. The most used strategy to validate the performance of the injury prediction is the cross-validation [19,24], even though it could suffer from an over-fitting problem. This problem could be caused by the fact that similar examples could be included in both training and test sets. In fact, this is usually induced by an overlapping of the training and test examples induced by the data preprocessing. For example, we could insert in the training set an example of day *n* and in the test set a training of the day *n* + 1. In this case, external workload features aggregate in acute (mean of the previous week), chronic (mean of the previous month), or ACWR strategies are created on almost all the same previous training examples. This aspect permits at the machine learning model to easily predict the injury risk due to the fact that an almost identical example was used to train it. Simulating the evolution of the competitive season permits for a better assessment of the accuracy of the injury prediction models, even if a similar problem could be detected. However, in contrast to the cross-validation approach, an evolutive scenario replicated what happened in the real world and consequently, the overfitting problem could be considered marginal. Only few papers simulate the evolutive scenario in order to validate their models [15,22,25]. In these papers, the train set was defined until day *n*, and the test set from day *n + i* simulating what happens as the season goes by.

Finally, one of the most important issues detected in the literature is that almost all of the papers do not compare their predictive performance results with a baseline model one. The baseline is a dummy classifier that makes predictions by using simple rules (e.g., always predict no injury class, always predict injury class, and stratified model in accordance with the injury examples distribution). This base model is useful for assessing if the machine learning model is really able to discriminate among players with different injury risks. In particular, if the trained model shows an accuracy similar to or lower than the baseline models, the machine learning algorithm could not be considered valid for injuries prediction.

Due to the great confusion regarding the application of machine learning techniques in sport science, the aim of this narrative review is to provide a guideline that permits to correctly build and evaluate injury prediction models. In particular, we describe in depth the strengths and limitations of each aspect needed to create a framework of big data analytic for injury forecasting. This paper describes all the features that could be used to predict injuries and all the possible preprocessing approaches (Section 2), how to train and test the predictive models (Section 4), and how to extract insights from interpretable and black box models (Section 5). Moreover, Figure 1 provides a schema of the framework to develop and test an injury prediction model.

## 2. Data Description

Several types of data were assessed to predict injuries. As provided in Figure 1 (pink leaves), two main categories were used as input features to predict injuries: (i) Training workload features, and (ii) players’ psychophysiological assessment features. Moreover, the red leaf in Figure 1 provides the label for the injury prediction model. The label refers to the output variable that the machine learning models try to predict.

In this section, we describe in depth all of these features (i.e., input and label features). and we also provide data preprocessing approach that could be useful to quantify the players’ history. Finally, we describe how to set the injury label to better train the machine learning models.

### 2.1. Input Features

#### 2.1.1. Training Workloads

It is possible to define two different types of training workloads: external, and internal (Figure 1).

##### External Workload

External workloads are defined as the training features that describe the effort performed during training or match sessions. Global Position System (GPS) commonly records such features. It is currently the most used system to track the movement of the players reporting information about three main workload features types [15]:kinematic: player’s overall movement during a training session, e.g., total distance and high-speed running distance (Distance in meters covered above 5.5 m/s);metabolic: energy expenditure of a player’s overall movement during a training session, e.g., high metabolic load distance (distance in meters covered by a player with a Metabolic Power is above 25.5 W/Kg);mechanical: player’s overall muscular-scheletrical load during a training session, e.g., explosive distance (Distance in meters covered above 25.5 W/Kg and below 19.8 Km/h), and the number of accelerations and decelerations above 2 and 3 m/s^2^.

These features are the most used to evaluate external workloads and to predict the risk of injury [19]. The accuracy of the data is affected by the GPS sampling rate. The most used GPS devices in soccer vary between 5 and 20 Hz, resulting in an error rate during acceleration, deceleration, and constant motion of approximately 3.1–11.3% and 1.9–6.0%, respectively [26]. Moreover, 1 Hz and 5 Hz GPS devices also show a consistent error when estimating distance during high-intensity running, velocity measures, and short linear running (particularly when this involves changes of direction) [26]. These errors could induce a possible under-/over- estimation of the external workloads. However, the higher the GPS sample rate, the higher the accuracy of the workloads metrics is. Despite this limitation, GPS seems to provide reliable data during team sports movements. In particular, 10 Hz and 15 Hz GPSs seem to be the most valid and reliable devices among linear and team sport simulated running [27]. Finally, one of the most important issues in metrics derived from GPS devices is that they provide many external workload features that are often related to each other. This aspect could result in a problem of multicollinearity that could negatively affect the prediction ability of the machine learning model. The accurate selection of the external workload features is a critical aspect not only for injury forecasting models, but also for every machine learning model. Hence, a feature selection process is required to solve this issue. This approach permits to diminish the number of input variables to reduce the computational cost of modeling, improve the performance of the model, and simplify the model interpretation. This approach is described in more depth below in the Section regarding Feature Selection.

##### Internal Workload

Internal training workload is usually assessed by evaluating the players’ Rate of Perceived Exertion (RPE). RPE is a self-reported scale describing the effort perceived by a player during a physical activity [28]. Two different RPE scales are used in sports: (i) CR-10 where the RPE values are ranged between 0 (no exertion at all) and 10 (maximal exertion), and (ii) 6–20 scale where the values are ranged between 6 (no exertion at all) and 20 (maximal exertion). The product between RPE (CR-10) and the total time of the training/match session is called Training Load (TL) [29]. TL is widely used in sports as an easy index describing the players’ internal workload. In particular, previous studies showed that TL is a valid method to estimate internal workload, highlighting the fact that this index is strictly related to the players’ physiological responses to training stimuli (e.g., oxygen consumption and heart rate) [29] and to external workloads of previous days that are found to be linked to players’ fatigue [30].

Another important feature describing the internal workload is heart rate (HR). Even if HR is an important objective index of internal load, the use of heart rate monitoring is not a standardized procedure in soccer teams due to the fact that the chest strap is uncomfortable while performing contact sports [31]. HR data recorded during training and matches are found to be strictly linked to energy expenditure, oxygen consumption, and maximal oxygen consumption, and consequently to internal workload [31,32]. However, even if HR accurately reflects the cardiovascular response to external stimuli, no paper has included this index in injury prediction tasks, probably due to the difficulty in recording it during training and competitions.

#### 2.1.2. Psycho-Physiological Assessment

Psycho-physiological data includes body composition information, cardiopulmonary assessments, injury history, neuromuscular assessments, players’ profile, players’ statistics, and psychological assessments [33]. In fact, several studies have found that individuals’ factors such as increased age [34], career duration [34,35], and previous injury [36,37] are related to the risk of injuries. Additionally, joint mechanical instability, general joint laxity, or functional instability are others factors affecting players’ injury risk [33,34]. Similarly, lack of training [35] or playing on a hard surface [33,38] also appear to increase the injury risk. Therefore, the combination of training workloads and psycho-physiological features allows for a complete overview of the players’ health status that might increase the accuracy of the machine learning models on injury prediction. In this section, we will provide a few examples of individual characteristics that could be used as an input variable for the injury prediction models.

##### Player Profile, Body Composition and Physical Assessment

Age, body weight, height, and the Body Mass Index (BMI) are common players’ characteristics used to develop models for injury prediction [15,20,21,22,23,24,25]. Moreover, human morphology—i.e., players’ bioimpedance parameters in both whole body and regional dimensions—could affect players’ injury risk. It is well known that morphology features (e.g., skeletal muscle mass) are related to force, power, sports performance, and oxygen uptake [39,40,41], and they are usually used to monitor rehabilitation phases [42]. These factors could be used to profile players in order to personalize the injuries prediction rules.

Additionally, physical tests (e.g., power, strength, and resistance) are widely used in soccer, and sports in general, to assess athletes’ physical status. Jump, sprint on 10 and 20 m, Change Of Direction (COD), and YoYo tests are only a few examples of all the tests performed on sport fields [43,44]. Moreover, neuromuscular assessment (e.g., core stability, joints range of motion, isokinetic joints flexion and extension strength, and Functional Movement Screen) are other evaluation tests that allow for the assessment the stability and mobility of the joints that are found to be linked to the injury risk [20,23,45,46].

The recent literature validates the use of infrared thermography for injury prevention due to its ability to detect skeletal muscle overload and fatigue in athletes [47]. After physical activity, athletes are exposed to physical stress that induces a change in blood flow profusion influencing skin temperature. The thermal asymmetry was found to be a potential index of injury risk related to training overload [47,48]. This non-invasive and fast technique might be every day used to assess the athlete’s muscle status. Moreover, creatine phosphokinase (CPK) and interleukin 6 (IL6) are two blood serum markers that indicate acute muscle damage and consequently provide a risk of injury [49,50]. Physical activity induces muscle damage that is accompanied by transient muscle strength loss, and an increase of muscle soreness [49,50,51]. Hence, these markers could help to better understand the muscle response to external stimuli helping to assess the injury risk.

Due to the difficulty of recording some of this information on a daily routine, they could be used to profile players over a specific time span in order to assess changes in individual characteristics that could affect injury risk. In this section, we have provided only a few examples of all the features that could be collected and inserted as input features in an injury forecaster.

##### Injury History

The players’ injury history is an important feature for injury forecasting [52]. The number of previous injuries is found to be related to the risk of injuries [15,53]. As a matter of fact, in injury forecasting models provided in recent papers, previous injury has been identified as one of the primary risk factors affecting injuries [15,20,21,23]. As a matter of fact, re-injuries of joints or muscles have been found to be induced by the fact that the players are not fully structurally and/or functionally restored from previous injuries [54]. In addition to the mere number of the previous injuries, Rossi and colleagues [15] highlight the fact that the distance to the day when the players return to the normal training routine is a factor that helps to predict injuries. In particular, in this study, the authors found that the players that are recently returned to regular training routines are more likely to incur in a new injury.

##### Self-Reported Wellness

Currently, the use of non-invasive and low-cost tools to monitor players’ status has become a suitable method of assessing players’ response to training and matches stimuli. For example, Perri and colleagues [55] investigate the relationship between wellness and Internal Training Load in Soccer detecting that the internal training workload is strongly related to the players’ wellness status of the next day. Moreover, other studies show that self-reported muscle soreness, sleep duration, sleep quality, and general wellness are also sensitive to TL variations in both Australian Football, English Premier Leagues and volleyball players [56,57,58].

Several questionnaires are used in sports to evaluate players’ well-being:Wellness Questionnaire (WELQUE) [59,60]: a 5-point Likert scale of 5 items (i.e., fatigue, sleep, soreness, stress, and mood), where 1 and 5 indicated the highest and lowest values of wellness for each item, respectively.Profile of Mood States (POMS) [61]: the primary psychological tool for monitoring training stress and over-training syndrome. The original questionnaire is composed of 65 items, but a short version with a subset of questions (i.e., tension-anxiety, depression-dejection, anger-hostility, fatigue-inertia, vigour-activity, and confusion-bewilderment) are widely used for assessing mood states among athletes [62]. For this inventory, the athletes report a 5-points Likert scale to rate how strongly they agreed with a statement.Daily Analyses of Life Demands for Athletes (DALDA) [63]: a self-reported questionnaire used to assess life-stress and symptoms of stress in athlete’s response to training. DALDA is divided into two sections: (i) self-assessment concerning the general stress sources that occur in the everyday life of an athlete, and (ii) determine what stress-reaction symptoms physically exist in the athlete.Total Quality Recovery (TQR) [64]: a self-reported scale ranged between 6 and 20 in order to evaluate the self-reported recovery status from a previous effort. Values of about 6 refer to no recovery at all, while 20 means that the athlete fully recover.Recovery-Stress Questionnaire for Athletes (RESTQ) [65]: this measures the frequency of current stress symptoms along with the frequency of recovery-associated activities. seven stress scales and five recovery scales characterized the RESTQ version for athletes.

##### Wearable Devices

Thanks to the technological revolution of the last decade, the use of wrist-worn devices (e.g., Apple Watch, Fit Bit, Garmin, and Polar) equipped with heart rate sensors are becoming extremely popular in sports [66]. These devices allow for a complete and objective overview of the users’ health status by passively recording semi-continuous heart rate, heart rate variability, blood oxygenation, steps, sedentary behaviours, and sleep quality. The recent literature validate the data provided by wrist-worn devices, establishing the profile error of the heart rate (HR) and heart rate variability (HRV) measurements [67,68,69,70,71]. Therefore, these devices allow for the objective and passive evaluate the well-being and recovery status of players in order to produce a potential index of injury risk. However, even if the use of this monitoring technology permits runners to run further and more frequently than those who did not use monitoring technologies, the use of these devices does not appear to be beneficial for injury incidence reduction [72].

#### 2.1.3. Data Preprocessing

Data preprocessing allows for aggregatation of time series data in order to provide more details about players’ training workloads history. In particular, several approaches have been used in sports and data sciences to aggregate individuals’ workloads:acute values reflect the mean value of the last week (from five to seven days);chronic values reflect the mean value of the past month (from 28 to 30 days);acute chronic workload ratio (ACWR) is the ratio between the acute and chronic values. ACWR values higher than 1 indicates that the acute values are higher than the chronic one, while vice versa for ACWR values lower than 1;monotony is the ratio between the mean value and the standard deviation of the training load during the past seven days;strain is the sum of the training loads for all training sessions during the past seven days multiplied by the monotony index.

The time series aggregation approach to compute these individuals’ features is referred to as the rolling mean or moving average. Such an approach consists of computing the mean value at the current day *t* of the past *n* days (*n* is the number of days that was set to compute the rolling mean, e.g., if *n* = 7 we are computing the mean of the past week). Moreover, the exponential weighted moving average (EWMA) is a type of rolling mean that permits the placing of a greater weight and significance on the most recent data, as shown in Equation (1). In this equation α refers to the specific decay, *y_t_* is the value at a time period *t* and *s_T_* refers to the value of the EWMA at any time period *t*:(1)$$s_{t}=\{\begin{array}{cc} y_{t}, & \;if\;t=1\\ \alpha *y_{t}+\left(1-\alpha \right)*s_{t}-1, & \;x\geq 0 \end{array}$$

The weight *α* could be set in three main different way: (i) *SPAN* (Equation (2)); (ii) *HALFLIFE* (Equation (3)); or (iii) Center Of Mass (*COM*, Equation (4)).
(2)$$\alpha_{SPAN}=\frac{2}{SPAN+1}$$
(3)$$\alpha_{HALFLIFE}=1-exp\left(\frac{\log\left(0.5\right)}{HALFLIFE}\right)$$
(4)$$\alpha_{COM}=\frac{1}{COM+1}$$

Murray et al. [11] suggest that the use of EWMA weighted by *α_SPAN_* is more effective in assessing the Australian football players’ injury risk. However, the rolling mean (not weighted) is a common approach for assessing the athletes’ training workloads history and, consequently, the risk of injury [15,22,73]. To the best of our knowledge, no study investigates the effect of the three different weights on training workloads assessment on injury risk. Future works are needed in order to solve this gap.

### 2.2. Target Feature

An athlete is commonly defined as injured when they are absent in physical activities for at least the day after the day of the onset (time-loss definition) [6,15,20,21,23,73]. Injuries could be classified into two main areas, i.e., contact and non-contact (soft-tissue) injuries. The vast majority of the papers in the literature aim to predict non-contact injuries. The main issue for predicting contact injuries is the fact that external events (e.g., kick from other players) are unpredictable and are not related to external or internal training workloads. In contrast, as asserted by Gabbett and colleagues in their paper, any illness related to training load (soft-tissue injuries) are commonly viewed as preventable [9].

One of the most important issues in injury modelling is the approach used for labelling each example. Labelling is the process that assigns class 0 (no-injury) or 1 (injury) to each feature vector example. In fact, we could predict injuries in the following day or in a specific time window. For example, we could predict the exact day when an injury occurs, or we could predict injuries that will happen within the next three or seven days. To predict injuries on day *n*, we should associate the injury label on the day *n* − *1*. This is due to the fact that on day *n* the players probably stop their physical activity before the end of the training or match sessions, making the prediction easier and consequently useless. Predicting injuries in day *n* (the day when the injury occurred) is equal to detecting it when the players are already injured. For this latter reason, the day when the injury occurred should be removed from the dataset, and the prediction should be made on day *n* − 1. Similarly, if we want to predict injuries in the next time window *I* (e.g., 3 days, 7 days, or 1 month), the injury label has to be set from day *n* − 1 − *i* to *n* − 1, and the injury day should be deleted.

## 3. Models

The orange leaves in Figure 1 provide the models that we could train and test for injury prediction. The algorithm is split into two main categories: (i) models referring to the “real” algorithms, and (ii) baseline models referring to “fake” (dummy) models and models provided in previous papers. The baseline models permit to assess the validity of the model proposed (see Section 3.2 Baseline for more details).

### 3.1. Machine Learning Models

Supervised machine learning models can learn a function that map an input (e.g., external workloads, internal workloads, and self-reported wellness) to an output (e.g., injury label). Van Eetvelde et al. [74], in their review about injury forecasting by machine learning model, asserted that the results detected in the previous papers are promising in the sense that these models might help coaches, physical trainers, and medical practitioners in the decision-making process for injury prevention and prediction. The most common supervised machine learning models used for injury forecasting are decision trees [15,20,23,24], binary logistic regression [6,11,15,24], random forests [15,22,24], and supporting vector machines [22,24]. Carrey et al. [22] also tested generalised-estimating equations, which are an extension of generalised linear models that account for correlations between repeated observations taken from the same subjects. Furthermore, Lopéz-Valenciano [20] and Ayala et al. [23] provided results from the AdaBoost classifier. Moreover, Vallance et al. [24] also trained K-Nearest Neighbours, Linear Discriminant Analysis, Ridge classifier, Gaussian Naive Bayes classifier, multi-layer perceptron and eXtreme gradient boosting algorithms. Additionally, Pappalardo et al. [75] proposed a convolutional neural network on multivariate time series. Finally, ensembled approaches that use multiple learning algorithms to obtain better predictive performance than could be obtained from any of the constituent learning algorithms alone were proposed by Lopéz-Valenciano et al. [20] and Ayala et al. [23].

### 3.2. Baseline

Validating the goodness of the injury prediction model is extremely important for comparing the prediction performance of the own model with different baseline models (dummy classifiers) and with the machine learning approaches provided in previous papers (more details about the prediction metrics for evaluating the model goodness are provided in Section 6 regarding prediction goodness). Similar performance results among the own model and the dummy models (models that make predictions without trying to find patterns in the data as described below) suggest that the own framework of data analytic is not able to detect patterns in data for an accurate prediction.

Only two studies [15,24] provide results of baseline models, validating the fact that the machine learning models are useful for detecting players that will get injured in the next training or matches. A dummy classifier is a model that makes predictions using naive rules. This classifier is useful as a simple baseline to compare with other (“real”) classifiers. Different strategies could be used to generate the dummy predictions:stratified: generates predictions by respecting the training set’s class distribution;most frequent: always predicts the most frequent label in the training set;prior: always predicts the class that maximizes the class prior (like most frequent strategy);uniform: generates ns uniformly at random;constant: always predicts a constant label that is provided by the user. This is useful for metrics that evaluate a non-majority class.

In addition to the dummy classifier, the comparison of the machine learning model performance goodness with the one of the models proposed by previous researchers is an important task for validating the prediction ability of the own model. Similar or lower predictive goodness results of the own framework of data analytics compared to the ones provided in the literature suggest that no improvement was brought to what is already present in the literature. Otherwise, higher predictive performance suggests that the own framework of data analytics provided better detected pattern into data to solve the predictive task.

Hence, the evaluation of the performance goodness of the own model compared with dummy ones and the ones provided by the previous paper is a mandatory task to misunderstand.

## 4. Train and Test

### 4.1. Validation

In order to assess the goodness of the machine learning model, the dataset was usually split into three different datasets. First of all, the model is initially fit on a training dataset, which is a set of examples used to fit the parameters. Successively, the model prediction goodness is validated one a validation set while tuning the model’s hyper-parameters. Finally, the test set is a dataset used to provide an unbiased evaluation of the final model.

The most common approach used to validate the injury forecasting in previous papers is the 10-fold cross-validation approach [20,21,22,23,24,73]. In contrast, Rossi et al. [15] proposed an interesting validation approach simulating the scenario that a soccer club usually incurs during the competitive season. In the cross-validation approach, the dataset is repeatedly split into training and validation sets. An additional test set, which should holds out from cross-validation, is required for validating the model’s performance. In order to be noticed in the cross-validation approach, a stratified split method is required to balance the distribution of the classes between folds (i.e., the distribution of the classes in each fold have to follow the one found in the entire dataset). To the best of our knowledge, this last step (evaluation of the predictive ability on the test set) is performed only in two previous works [15,22]. Rossi et al. [15], trained and validated their models by week *n*, while testing them in the week *n* + 1. In contrast, Carry et al. [22] trained and validated their models in season, and tested the fitted model in the following season. Therefore, almost all of the models proposed in the literature are not yet tested; for this reason, future works are required for assessing the real goodness of the injury forecasting. By using an evolutive scenario, as proposed in [15], it is possible to test the ability of the classifiers to predict injuries in new data that are not already present in the dataset where the machine learning models are trained and validated.

The main strength of the evolutive scenario is the possibility to simulate what will happen during the soccer season, allowing for a deeper test of the model. Moreover, in contrast to the approach proposed by Carrey et al. [22], by using the validation approach proposed by Rossi et al. [15] it is possible to re-train models inserting new injury and no-injury examples as the season goes by increasing the sample size as the season continues. Let us assume that a soccer club has data until day *n,* and it wants to detect the risk of injuries of the next days *i* (between *n* + 1 and *n + i*). The machine learning models are trained and validated (cross-validated) over the training set (i.e., until day) and then tested by predicting the injury class (no-injury an injury) by using new data recorded in day *n* + 1, *n* + 2, …, *n + i*. This test approach is mandatory for researchers that are interested in evaluating the predictive performance of their model in a real scenario where new data are generated every day.

### 4.2. Data Processing for Each Training Fold

A data preprocessing phase is required for each training fold in both cross-validation and evolutive scenario approaches. First of all, a sampling approach is useful for balancing the target classes, due to the fact that the number of injury examples is enormously lower than the no-injury ones. Moreover, to reduce features space and consequently improve the interpretability of the machine learning model, a feature selection process permits selecting the best features to predict injuries (Section 4.2.2 Feature Selection).

#### 4.2.1. Sampling

Over- and under- sampling in data analysis are techniques to adjust the class distribution of a dataset. Oversampling permits to increase in the sample size of the minority class, while under-sampling reduces the number of examples of the majority class to balance the two classes. Due to the fact that the injury dataset usually shows only 2% of injury examples over the total number of observations, an over-sampling approach is needed to balance the two classes. Synthetic minority over-sampling technique (SMOTE) is the most common approach employed in the previous study, allowing for an increase in the injury prediction goodness compared to models trained without oversampling process [15,20,22,23]. This approach considers k nearest neighbours of a current data point in the multidimensional feature space. To create a synthetic data point, SMOTE algorithm takes the vector of the difference between one of those k neighbours and the current data point. Then, it multiplies this vector by a random number ranging between 0 and 1, and adds this vector to the current data point to create the new synthetic examples. Such an oversampling approach used in some previous works increased the injury prediction performance [20,22,23]. In contrast, Rossi et al. [15] used a modified version of SMOTE algorithm called adaptive synthetic sampling approach (ADASYN). This approach derives from the SMOTE algorithm, showing just one main difference. ADASYN will select a particular point for duping towards points that are located not in homogeneous neighbourhoods. Moreover, Carey and colleagues [22] compare the effect of the sampling approach to injury prediction detecting that under-sampling non-injury class decreases the model performance, while SMOTE does not lead to any performance improvement. In that paper, the authors asserted that the under-sampling approach might not be appropriate for injury forecasting due to the fact that there is the possibility of lost information by removing a large number of non-injury examples. Furthermore, SMOTE could be useful for detecting injury patterns, but may be useless if new injuries show little resemblance to the past ones. However, not engaging in any type of oversampling approach could lead to the same problem because new injuries could differ from the previous ones and consequently could not be possible to predict.

It is worth noting that the oversampling process should be performed only on the training set in order to avoid any possible risk of overfitting. In fact, the examples created by oversampling techniques are based on similarity of injury examples in the dataset and, consequently, it will create “fake” examples that are similar to the real ones. Hence, if such synthetic examples are created based on the entire dataset (before performing the training and testing split), we will use similar data in both train and test sets resulting in a problem of overfitting during validation.

#### 4.2.2. Feature Selection

Feature selection is the process of selecting a subset of the most relevant features that are linked to the injury prediction output. This technique reduces the dimensions of the feature space, making the machine learning models easier to interpret and faster to train, with a lower risk of multi co-linearity issues [76]. To the best of our knowledge, only one study used such a technique. Rossi et al. [15] selected the best features with reference to injury forecasting by using a Recursive Feature Elimination with Cross-validation (RFECV) process. The goal of RFE is to select features by recursively considering smaller and smaller sets of features. First, the estimator is trained on the entire train dataset and the importance of each feature is obtained. Then, the least important features are deleted from the current set of features, and this procedure is recursively repeated on the subset of features until the best number of features are selected. Cross-validation is useful for selecting the features by RFE in a different subset of the training dataset in order to select the most important and the most constant features obtained in each fold. Feature selection should be performed on the training set in order to check for eventual overfitting issues during model goodness assessment.

### 4.3. Hyper-Parameters Fit on Validation Set

In the validation set, is it possible to fit the best machine learning hyper-parameters to improve the injuries prediction performances. Hyper-parameter tuning is crucial, as they control the overall behaviour of a machine learning model. Every machine learning model has different hyper-parameters that can be set. One traditional and popular way to perform hyper-parameter tuning is called Grid Search. However, only two papers were used to fit the hyper-parameters to improve the performance of their injury forecasting models [15,22]. This method tries every possible combination of hyper-parameters input values. Using this method, we can find the best set of hyper-parameters values for each model in the parameter search space. This usually uses more computational resources. Another approach to detecting the best hyper-parameters is the Random Search. The only difference compared to the grid search approach is that it tries random combinations of hyper-parameters instead of trying every possible combination. Due to this, the latter approach can be extremely effective in practice due to the lower computational time required, but there is no guarantee that the same optimal result as Grid Search will be obtained.

## 5. Model Interpretation

Currently, explainable artificial intelligence is a hot topic in data science. Understanding the decision-making process allows for more confidence in the model’s prediction. In particular, the model interpretation allows for the investigation of the reasons behind the predicted injuries, and consequently allows coaches and athletic trainers to modify the training schedule reducing the risk of injury maximizing the effect of training.

Interpretable models such as decision trees allow for the extraction of rules that describe the reason why the classifier assigns a player to injury or no-injury classes. In contrast, black-box models such as Neural Network, which generally produces more accurate models compared to interpretable ones, do not permit to easier interpret the decision-making process. Pappalardo et al. [75] proposed a method to explain their convolutional neural network model by using the SHAP (Shapley Additive exPlanations) method. SHAP uses game theory to explain the output of any machine learning model. Given a pre-trained classifier and an example to classify for each input feature, SHAP explainer computes a value that represents how much that feature has influenced the classifier’s decision-making process by providing global and local explanations of the machine learning models.

## 6. Prediction Goodness

Confusion matrix, also called error matrix, is a specific contingency table (with two dimensions, i.e., “actual” and “predicted”), that allows for the visualization of the performance of an algorithm (see Table 1). In this table, each row represents the instances in a predicted class, while each column represents the instances in an actual class.

Several metrics are derived from the confusion matrix (Table 1) in order to evaluate machine learning classifiers goodness (green leafs in Figure 1):Precision (Sensitivity) is the ratio between the true positive (TP) and all the positive results, i.e., the sum of the true positive and the false positive (FP): TP/(TP + FP). The positive class in this example is the injury one. Precision indicates the fraction of examples that the classifier correctly classifies in a given class over the number of all examples the classifier assigns to that class.Recall (Specificity) is the ratio between TP and the number of all samples that should have been identified as positive, i.e., the sum of TP and False Negative (FN): TP/(TP + FN). The recall an index that indicates the number of examples that a classifier correctly classified in a given class.F1-score = 2(precision × recall)/(precision + recall). This is the harmonic mean of Sensitivity (precision) and Specificity (recall).Area Under the Curve (AUC) is an aggregate measure of performance across all possible classification thresholds. In particular, it is the probability that a model ranks a random positive instance is higher than a randomly negative one. The AUC score is ranged between 0.5 and 1. The higher the AUC, the higher the accuracy of the model.

It should be noted that assessing the performance of the classifiers as the metrics mean of the two classes is not the right choice for assessing the injury prediction models’ goodness due to the fact that the dataset is extremely unbalanced, resulting in an average value strongly affect by no-injury class. Hence, assessing the metric values in both the classes allows for a better understanding of the model goodness. In particular, the injury class metric values provide the ability of the machine learning models for injury prediction.

In a cross-validation approach, the mean and the standard deviation of the metrics values detected for each fold provide the goodness of the models and their variability. The higher is the standard deviation the lower is the stability of the model, i.e., the data in the trained model higher affect the learning ability of the machine learning model. In contrast, in an evolutive scenario, it is possible to assess the goodness of the machine learning algorithms day-by-day or week by week. Moreover, the cumulative performance of the classifiers permits to assess its goodness from the first day of prediction in the evolutive scenario [15].

## 7. Conclusions

This narrative review could help data and sports scientists to develop machine learning models in sports. Even if the framework of data analytic proposed in this paper was focused on injury forecasting, it is applicable in all sports topics. The results derived from the application of machine learning models could assist coaches and sports managers in result prediction, athlete performance assessment, sports talent identification, game strategy evaluation, and other topics. For example, Pappalardo et al. [77] evaluated the performance of soccer players by using a data-driven approach. In this study, the authors trained a machine learning model to predict the results of a match in order to extract the weights that each event (e.g., number of passes, number of shots, and number of tackles) performed by a soccer player during the match affect to win the game. Based on these weights, the authors created a score by a scalar product between the previously computed feature weights and the values these features achieve in that match played by that player. Moreover, Salabun and colleagues [78] developed a multi-criteria model (i.e., Characteristic Objects Method—COMET) to evaluate players’ performance in accordance with their position on the field based on their match statistics. Additionally, the COMET approach was also proposed by a few papers to assess the progression of swimmers over the competitive season showing a good accuracy compared to other models previously proposed in the scientific literature [79,80].

In this paper, a guideline that allows for correctly building and evaluating injury prediction models was provided. It was highlighted how to preprocess the data, how to correctly train, validate, and test the predictive models, how to extract insight from withe and black-box models, and how to evaluate the prediction goodness. The standardization of the learning approach will allow for a comparison of results from different papers, reducing the risk of misleading results interpretation. The development of a robust prediction model for any of the possible sports topics could allow clinical and field experts to better estimate probabilities of which informed decisions could occur, providing advantages in player health and sport management.

## Figures and Tables

**Figure 1 sports-10-00005-f001:**
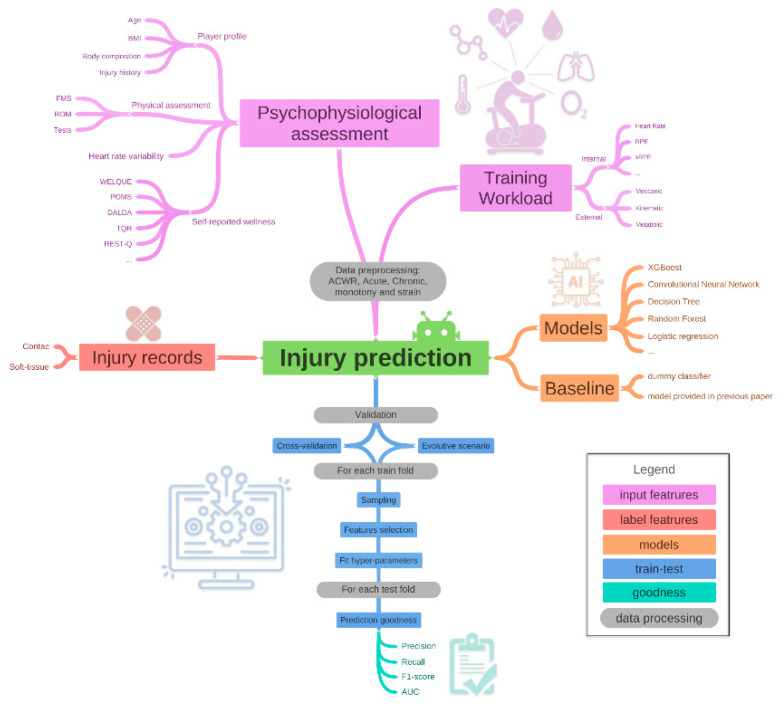
Diagram of the injury forecasting validation. The pink leaves (i.e., psychophysiological assessment and training workload) refer to the input variables for the injury prediction algorithm. The red leaf is injury information used to label each training vector. Orange leaves are the models trained and tested by the injury prediction algorithm. Each of these three leaf types (pink, red, and orange) are useful for building the injury prediction algorithm. Furthermore, blue leaves describe how to train, validate, and test the model developed by the injury prediction algorithm. Moreover, green leaves list all the metrics to assess the model’s goodness. Finally, gray leaves describe the data preprocessing in each injury prediction algorithm stage.

**Table 1 sports-10-00005-t001:** Confusion matrix. TP, FP, TN, and FN refer to True Positive, False Positive, True Negative, and False Negative, respectively.

		Actual Classes	
		Injury	No-Injury
Predicted classes	Injury	TP	FP
	No-Injury	FN	TN

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
