# Peer review of "A Narrative Review for a Machine Learning Application in Sports: An Example Based on Injury Forecasting in Soccer"

_sports, 2021, doi:10.3390/sports10010005_

Round 1

Reviewer 1 Report

A machine learning algorithm for injury forecasting is a very important subject that has quickly increased. The authors present state-of-the-art machine-learning techniques in sport science. The paper seems to be interesting, however, some shortcomings must be eliminated. The list of my comments is as follows:
1. In my opinion, the authors should eliminate the word: 'Handbook'. This is state-of-the-art. This is not a handbook.  
2. The abstract should be extended. There should be more emphasis contribution of this work.
3. Methodology must be more carefully described.
4. There is a lack of the papers about swimmers, e.g., Swimming progression evaluation by assessment model based on the COMET method; A fuzzy assessment model for freestyle swimmers-a comparative analysis of the MCDA methods; or other sports A Fuzzy Inference System for Players Evaluation in Multi-Player Sports: The Football Study Case
5. There should be also shown in general how ML is used in support sport decisions (not only injury).
6. The conclusions must be extended - please add future research directions.

Author Response

Reviewer 1 general comment. A machine learning algorithm for injury forecasting is a very important subject that has quickly increased. The authors present state-of-the-art machine-learning techniques in sport science. The paper seems to be interesting, however, some shortcomings must be eliminated.

Authors’ answer: We would like to thank the reviewer for their useful comments and suggestions. We have answered point-by-point at all of their comments below.

Point 1.1. In my opinion, the authors should eliminate the word: 'Handbook'. This is state-of-the-art. This is not a handbook.  

Authors’ answer: We agree with the reviewer’s comment. We have modified the title as “A narrative review for a machine learning application in sports: an example based on injury forecasting in soccer”.

Point 1.2. The abstract should be extended. There should be more emphasis on the contribution of this work.

Authors’ answer: thanks for this useful suggestion. We have edited the abstract in order to give more emphasis on the contribution of this work. The abstract was changed as “In the last decade, the number of studies about machine learning algorithms applied to sports, e.g. injury forecasting and athlete performance prediction, have quickly increased. Due to the number of works and experiments already present in the state-of-the-art regarding machine-learning techniques in sport science, the aim of this narrative review is to provide a guideline describing a correct approach to train, validate and test machine learning models to predict events in sports science. The main contribution of this narrative review is to highlight any possible strengths and limitations during all the stages of model development, i.e. train, validation, test, and interpretation, in order to limit possible errors that could induce misleading results. In particular, this paper shows an example about injury forecaster that provides a description of all the features that could be used to predict injuries, all the possible preprocessing approaches for time series analysis, how to correctly split the dataset to train and test the predictive models, and the importance to explain the decision-making approach of the white and black box models.”

Point 1.3. Methodology must be more carefully described.

Authors’ answer: In the narrative review, the literature review is usually not systematic and for this reason, the search methodologies are not reported in this kind of paper. For this reason, we decided not to add this point in the text. Additionally, we provided three examples of narrative reviews published on MDPI Sports journal that do not provide search methodology information:

-              https://www.mdpi.com/2075-4663/9/11/154/htm

-              https://www.mdpi.com/2075-4663/7/7/172/htm

-              https://www.mdpi.com/2075-4663/9/5/59/htm

Point 1.4. There is a lack of the papers about swimmers, e.g., Swimming progression evaluation by assessment model based on the COMET method; A fuzzy assessment model for freestyle swimmers-a comparative analysis of the MCDA methods; or other sports A Fuzzy Inference System for Players Evaluation in Multi-Player Sports: The Football Study Case

Authors’ answer: Due to the fact that the examples provided in this paper are mainly focused on injury prediction in team sport and in particular on soccer, the papers suggested by the reviewer are out of topic. However, we have tried to integrate examples of application of different model in the Conclusion section as: “…Moreover, Salabun and colleagues [78] developed a multi-criteria model (i.e., Characteristic Objects Method - COMET) to evaluate players' performance in accordance with their position on the field based on their match statistics. Additionally, the COMET approach was also proposed by a few papers to assess the progression of swimmers over the competitive season showing a good accuracy compared to other models previously proposed in the scientific literature [79,80].”

Point 1.5. There should also be shown in general how ML is used in support sport decisions (not only injury). The conclusions must be extended - please add future research directions.

Authors’ answer: We would like to thank the reviewer for these useful comments. We have extended the Conclusion section adding some examples of ML application on sports in the Conclusion section as:

“This narrative review could help data and sports scientists to develop machine learning models in sports. Even if the framework of data analytic proposed in this paper was focused on injury forecasting, it is applicable in all of the sports topics. Actually, the results derived from the application of machine learning models could assist coaches and sports managers in result prediction, athlete performance assessment, sports talent identification, game strategy evaluation and more other topics. For example, Pappalardo et al. [77] evaluated the performance of the soccer players by using a data-driven approach. In this study, the authors trained a machine learning model to predict the results of a match in order to extract the weights that each event (e.g., number of passes, number of shots, and number of tackles) performed by a soccer player during the match affect to win the game. Based on these weights the authors of this work create a score by a scalar product between the previously computed feature weights and the values these features get in that match played by that player. Moreover, Salabun and colleagues [78] developed a multi-criteria model (i.e., Characteristic Objects Method - COMET) to evaluate players' performance in accordance with their position on the field based on their match statistics. Additionally, the COMET approach was also proposed by a few papers to assess the progression of swimmers over the competitive season showing a good accuracy compared to other models previously proposed in the scientific literature [79,80].

In this paper, it was provided a guideline that permits to correctly build and evaluate injury prediction models. Actually, it was highlighted how to preprocess the data, how to correctly train, validate, and test the predictive models, how to extract insight from withe and black-box models, and how to evaluate the prediction goodness. The standardization of the learning approach will permit to compare results from different papers reducing the risk of misleading results interpretation. The development of a robust prediction model for any of the possible sports topics could allow clinical and field experts to better estimate probabilities of which informed decisions could occur providing advantages in player health and sport management.”

Reviewer 2 Report

Dear Authors,

The content of the manuscript is of great interest and relevance. However, it has some shortcomings, especially from the formal point of view, which should be corrected.

In the Introduction (and in Discussion) I believe it is necessary to mention some studies relevant to the subject matter covered:
doi: 10.3390/jcm10050902
doi: 10.1186/s13102-021-00363-4
doi: 10.1186/s13102-021-00347-4

The structure and subsections of the manuscript should be thoroughly reorganized to conform to the formal standards of scientific writing.

The conclusions are poor considering the great work done by the authors.

Kind regards

Author Response

Reviewer 2 general comment. The content of the manuscript is of great interest and relevance. However, it has some shortcomings, especially from the formal point of view, which should be corrected.

Authors’ answer: We would like to thank the reviewer for their useful comments and suggestions. We have answered point-by-point at all of their comments below.

Point 2.1. In the Introduction (and in Discussion) I believe it is necessary to mention some studies relevant to the subject matter covered:

doi: 10.3390/jcm10050902

doi: 10.1186/s13102-021-00363-4

doi: 10.1186/s13102-021-00347-4

Authors’ answer: Even if the paper suggested by the reviewer is a little bit out of the paper’s topic, (machine learning application based on an example of injury prediction models in soccer), we have tried to add some details about other sports such as:

“…Differently, a low number of injuries was detected in kayakers. Actually, only 1.1 injuries/year in men and 1.5 injuries/year in women was detected in this group of athletes [5]. The most common injuries in kayakers are on shoulders, followed by knees, lower back, and the other segments of the upper limbs [5].”

“… Hence, these devices permit to objectively and passively evaluate the well-being and recovery status of the players that could be an index of injury risk. However, even if the use of this monitoring technology permits to runners to run further and more frequently than those who did not use monitoring technologies, the use of these devices seems to not be beneficial on injury incidence reduction [72].”

Point 2.2. The structure and subsections of the manuscript should be thoroughly reorganized to conform to the formal standards of scientific writing.

Authors’ answer: A narrative review is a review of what is considered relevant for the topic and the aim of the review, but without a specified methodological plan as for a systematic review. Usually, this type of review does not follow a specific structure. MDPI guidelines do not provide any limitation for this type of paper and for the readability of the paper we would prefer not to change the structure of the text. Here are some examples of narrative review published in the MDPI Sports journal that do not follow any specific paper’s structure:

  • https://www.mdpi.com/2075-4663/9/11/154/htm
  • https://www.mdpi.com/2075-4663/7/7/172/htm
  • https://www.mdpi.com/2075-4663/9/5/59/htm

Point 2.3. The conclusions are poor considering the great work done by the authors.

Authors’ answer: We would like to thank the reviewer for this useful comment. We have extended the Conclusion section as:

“This narrative review could help data and sports scientists to develop machine learning models in sports. Even if the framework of data analytic proposed in this paper was focused on injury forecasting, it is applicable in all of the sports topics. Actually, the results derived from the application of machine learning models could assist coaches and sports managers in result prediction, athlete performance assessment, sports talent identification, game strategy evaluation and more other topics. For example, Pappalardo et al. [77] evaluated the performance of the soccer players by using a data-driven approach. In this study, the authors trained a machine learning model to predict the results of a match in order to extract the weights that each event (e.g., number of passes, number of shots, and number of tackles) performed by a soccer player during the match affect to win the game. Based on these weights the authors of this work create a score by a scalar product between the previously computed feature weights and the values these features get in that match played by that player. Moreover, Salabun and colleagues [78] developed a multi-criteria model (i.e., Characteristic Objects Method - COMET) to evaluate players' performance in accordance with their position on the field based on their match statistics. Additionally, the COMET approach was also proposed by a few papers to assess the progression of swimmers over the competitive season showing a good accuracy compared to other models previously proposed in the scientific literature [79,80].

In this paper, it was provided a guideline that permits to correctly build and evaluate injury prediction models. Actually, it was highlighted how to preprocess the data, how to correctly train, validate, and test the predictive models, how to extract insight from withe and black-box models, and how to evaluate the prediction goodness. The standardization of the learning approach will permit to compare results from different papers reducing the risk of misleading results interpretation. The development of a robust prediction model for any of the possible sports topics could allow clinical and field experts to better estimate probabilities of which informed decisions could occur providing advantages in player health and sport management.”

Round 2

Reviewer 1 Report

The paper has been improved and can be accepted in the present form. 

Reviewer 2 Report

First of all, I would like to thank you for the trust you have placed in me for the evaluation of this manuscript. It has been an honor for me.

Regarding the submitted manuscript. It seems to me that the improvements made by the authors have significantly improved the quality of the text.

In particular, the expansion of the Introduction and Conclusions is essential to give the article the necessary clinical impact. In addition, the reorganization of Materials & Methods facilitates the reproducibility of the research and makes it easier for future readers to read.